# Markers of NETosis in Patients with Systemic Lupus Erythematosus and Antiphospholipid Syndrome

**DOI:** 10.3390/ijms24119210

**Published:** 2023-05-24

**Authors:** Tatiana Reshetnyak, Kamila Nurbaeva, Ivan Ptashnik, Anna Kudriaeva, Alexey Belogurov, Aleksandr Lila, Evgeny Nasonov

**Affiliations:** 1V.A. Nasonova Research Institute of Rheumatology, 115522 Moscow, Russia; camila9@mail.ru (K.N.); amlila@mail.ru (A.L.); elnasonov@mail.ru (E.N.); 2Department of Rheumatology, The Russian Medical Academy of Continuing Professional Education, 125993 Moscow, Russia; 3Shemyakin-Ovchinnikov Institute of Bioorganic Chemistry of the Russian Academy of Sciences, 117997 Moscow, Russia; iv.ptash@gmail.com (I.P.); anna.kudriaeva@ibch.ru (A.K.); belogurov@ibch.ru (A.B.J.); 4Department of Biological Chemistry, Evdokimov Moscow State University of Medicine and Dentistry, Ministry of Health of Russian Federation, 127473 Moscow, Russia

**Keywords:** NETosis, MPO-DNA complex, nucleosomes, systemic lupus erythematosus, antiphospholipid syndrome, antiphospholipid antibodies, nephritis

## Abstract

Neutrophil Extracellular Traps (NETs) have been implicated in systemic lupus erythematosus (SLE) and antiphospholipid syndrome (APS) pathogenesis. The myeloperoxidase–deoxyribonucleic acid (MPO-DNA) complex and nucleosomes are serum markers of NETosis. The aim of this study was to assess these NETosis parameters as markers for SLE and APS diagnosis and their association with clinical features and disease activity. A total of 138 people were included in the cross-sectional study: 30 with SLE without APS, 47 with SLE and APS, 41 patients with primary antiphospholipid syndrome (PAPS), and 20 seemingly healthy individuals. Serum MPO-DNA complex and nucleosome levels were determined via an enzyme-linked immunosorbent assay (ELISA). Informed consent was obtained from all subjects involved in the study. The Ethics Committee of the V.A. Nasonova Research Institute of Rheumatology (Protocol No. 25 dated 23 December 2021) approved the study. In patients with SLE without APS, the levels of the MPO-DNA complex were significantly higher compared to patients with SLE with APS, with PAPS, and healthy controls (*p* < 0.0001). Among patients with a reliable diagnosis of SLE, 30 had positive values of the MPO-DNA complex, of whom 18 had SLE without APS, and 12 had SLE with APS. Patients with SLE and positive MPO-DNA complex levels were significantly more likely to have high SLE activity (χ^2^ = 5.25, *p* = 0.037), lupus glomerulonephritis (χ^2^ = 6.82, *p* = 0.009), positive antibodies to dsDNA (χ^2^ = 4.82, *p* = 0.036), and hypocomplementemia (χ^2^ = 6.72, *p* = 0.01). Elevated MPO-DNA levels were observed in 22 patients with APS: 12 with SLE with APS and 10 with PAPS. There were no significant associations between positive levels of the MPO-DNA complex and clinical and laboratory manifestations of APS. The concentration of nucleosomes was significantly lower in the group of SLE patients (±APS) compared to controls and PAPS (*p* < 0.0001). In SLE patients, the frequency of low nucleosome levels was associated with high SLE activity (χ^2^ = 13.4, *p* < 0.0001), lupus nephritis (χ^2^ = 4.1, *p* = 0.043), and arthritis (χ^2^ = 3.89, *p* = 0.048). An increase in the specific marker of NETosis, the MPO-DNA complex, was found in the blood serum of SLE patients without APS. Elevated levels of the MPO-DNA complex can be regarded as a promising biomarker of lupus nephritis, disease activity, and immunological disorders in SLE patients. Lower levels of nucleosomes were significantly associated with SLE (±APS). Low nucleosome levels were more common in patients with high SLE activity, lupus nephritis, and arthritis.

## 1. Introduction

Systemic lupus erythematosus (SLE) is an autoimmune rheumatic disease characterized by the hyperproduction of organ-nonspecific autoantibodies to the components of the nucleus and cytoplasm with the formation of immune complexes that cause immune-inflammatory damage to various organs [1]. Antiphospholipid syndrome (APS) is an autoimmune thrombophilia characterized by recurrent thrombosis, obstetric pathology, and the presence of persistently positive antiphospholipid antibodies (aPL) [2]. There is a combination of SLE and APS in up to 40% of cases, which indicates the common pathogenetic pathways of both diseases [3]. Neutrophil activation is considered to be an important mechanism in the development of both SLE and APS [4]. NETosis is a process of the formation of web-like structures referred to as neutrophil extracellular traps (NETs) by activated neutrophils [5]. It has been established that NETs have proinflammatory and prothrombogenic features [6]. Serum levels of NETs can be determined by assessing specific markers of NETosis, such as the myeloperoxidase–deoxyribonucleic acid complex (MPO-DNA), the neutrophil elastase-DNA complex, citrullinated histones, or nonspecific indicators such as nucleosomes and circulated cell-free deoxyribonucleic acids (cfDNA) [7]. The MPO-DNA complex is the most specific marker for determining the level of NETs in peripheral blood [8]. Nucleosomes are a structural part of chromatin formed by DNA and histone proteins, which, on the one hand, can be autoantigens in SLE [9], and, on the other hand, serve as markers of NETs [10]. There are limited data on the clinical significance of the determination of the MPO-DNA complex and nucleosomes in SLE and APS. The aim of this study was to assess these NETosis parameters as markers for SLE and APS diagnosis and their association with clinical features and disease activity.

## 2. Results

### 2.1. MPO-DNA Complex in the Studied Groups

The levels of the MPO-DNA complex significantly differed in the studied groups compared to controls (Figure 1). In patients with SLE without APS, the MPO-DNA complex was significantly higher compared to patients with SLE with APS, patients with PAPS, and healthy controls (HC). The levels of the MPO-DNA complex were comparable between patients with SLE and APS, patients with PAPS, and healthy controls.

To assess the ability of the MPO-DNA complex to differentiate patients with SLE without APS from healthy donors, ROC analysis was performed. According to ROC analysis, the AUC for the MPO-DNA complex was 0.876 (*p* < 0.0001). The value of the MPO-DNA complex at the cut-off value of 0.058 with a sensitivity of 80% and a specificity of 74.5% allows differentiating SLE patients without APS from healthy controls (Figure 2).

### 2.2. Clinical Characteristics of SLE Patients and the Serum MPO-DNA Complex Levels

The number of patients with SLE in the study, regardless of APS, was 77. The upper 95th percentile reference limit for the MPO-DNA complex was 0.09335 OD_450_. Depending on MPO-DNA complex positivity, all patients with SLE were divided into two groups: Group 1 (n = 30) had positive MPO-DNA complex values (>0.09335 OD_450_), and Group 2 (n = 47) had negative MPO-DNA complex values (≤0.09335 OD_450_). There were 18 patients with SLE without APS and 12 patients with SLE and APS in Group 1, and there were 12 patients with SLE without APS and 35 with SLE and APS in Group 2. The frequency of complex MPO-DNA detection was associated with SLE without APS. Elevated MPO-DNA complex levels were recorded in 18 of 30 (60%) patients with SLE without APS versus 12 (25.5%) of 47 SLE patients without APS and a negative MPO-DNA complex (*p* = 0.005).

Patients with positive levels of the MPO-DNA complex were significantly more likely to have high SLE activity, positive antibodies to dsDNA, hypocomplementemia, and lupus glomerulonephritis (Table 1). Other clinical manifestations were not significantly associated with MPO-DNA positivity.

### 2.3. The MPO-DNA Complex Levels Depend on the Disease Activity and Clinical Manifestations of SLE

SLE patients (n = 77) were divided into two groups depending on SLE activity according to SLEDAI-2K: group I (SLEDAI 2K ≥ 11 points) included 22 patients with high disease activity, and group II (SLEDAI 2K < 11 points) included 55 patients with moderate or low SLE activity. The MPO-DNA complex levels were significantly higher in patients of Group I with high disease activity compared with patients of Group II (Figure 3).

The MPO-DNA complex levels positively correlated with SLEDAI-2K and anti-dsDNA. The MPO-DNA complex levels negatively correlated with complement components C3 and C4 (Figure 4).

Patients with SLE and glomerulonephritis had significantly higher levels of the MPO-DNA complex (Figure 5). Other clinical manifestations of SLE had no significant effect on serum MPO-DNA complex levels.

The MPO-DNA complex levels were significantly higher in patients with elevated anti-dsDNA levels and hypocomplementemia (Table 2).

Therapy with glucocorticoids, DMARDs, or biologic DMARDs had no significant effect on the levels or frequency of MPO-DNA complex positivity.

### 2.4. MPO-DNA Complex in Antiphospholipid Syndrome

Elevated MPO-DNA complex levels were noted in 22 patients with APS: 12 (25.5%) of 47 patients with SLE with APS and 10 (23.4%) of 41 with PAPS. No significant associations between positive levels of the MPO-DNA complex and clinical and laboratory manifestations of APS were observed.

### 2.5. Nucleosomes in Patients with SLE and APS

The concentration of nucleosomes was significantly lower in the group of SLE patients without APS compared to controls and patients with PAPS. The nucleosome levels were also lower in the SLE with the APS group compared to controls and PAPS. Nucleosome levels in the group of patients with PAPS did not differ significantly from healthy controls (Figure 6).

### 2.6. Clinical Characteristics of SLE Patients and Serum Nucleosomes Levels

The lower 5th percentile reference limit for nucleosomes was 0.1528 OD_450_. Patients with SLE were divided into two groups: those with low levels of nucleosomes (n = 45) and those with normal values of nucleosomes in blood serum (n = 32). The frequency of low nucleosome levels was associated with lupus nephritis, arthritis, and SLE activity according to SLEDAI-2K (Table 3).

Patients with high SLE activity, lupus nephritis, and arthritis had significantly lower nucleosome levels (Figure 7). The presence of other clinical manifestations of SLE had no significant effect on nucleosome levels.

Nucleosome levels negatively correlated with SLEDAI-2K (Figure 8).

Nucleosome levels negatively correlated with the MPO-DNA complex levels in patients with SLE (Figure 9).

### 2.7. Nucleosomes in Antiphospholipid Syndrome

Low serum nucleosome levels were noted in 32 of 88 patients with APS (36.4%), including 24 (51.1%) of 47 patients with SLE with APS and 8 (19.5%) of 41 patients with PAPS. Patients with highly positive levels of aCL IgM and antibodies to β2GP1 IgM were significantly less likely to have decreased serum nucleosome levels (Table 4). Other clinical and laboratory manifestations of APS were not statistically significant, depending on nucleosome levels.

No correlation was found between nucleosomes and the MPO-DNA complex in APS (*p* = 0.63).

## 3. Discussion

Neutrophil extracellular traps are web-like structures consisting of decondensed chromatin and proteins of neutrophils’ granules, nucleus, and cytoplasm, such as neutrophil elastase, myeloperoxidase, cathepsin G, and other antimicrobial proteins [11]. In addition to the antimicrobial function, the pathological role of NETs in the development of many inflammatory diseases and thrombosis has been established [12]. Data on the relationship between NETs levels and clinical and laboratory manifestations of SLE and APS are contradictory.

The MPO-DNA complex is a specific marker of NETs that is predominantly formed during NETosis. Determination of the MPO-DNA complex is an interesting target for the evaluation of NETs because it has several advantages. For example, unlike citrullinated histones, it does not depend on the activation of the peptidylarginine deiminase 4 (PAD4) enzyme, which is required for histone citrullination, and unlike circulated cell-free DNA (cfDNA), it is not formed during necrosis or apoptosis [7].

In the present study, significantly higher levels of the MPO-DNA complex were found in patients with SLE without APS compared to healthy controls, which agrees with the data of Hanata et al. [13] and Bruschi et al. [14]. In addition, Bruschi et al. [14] demonstrated that the MPO-DNA complex was a diagnostically significant marker of SLE; they found that the AUC value for differentiating SLE from healthy controls was 0.820 for the MPO-DNA complex. In the present study, the AUC for the MPO-DNA complex for the diagnosis of SLE was also high (0.876) (Figure 2). In turn, other authors have studied the ability to form [15,16] and degrade NETs [17,18,19], as well as levels of nonspecific markers of NETs [20,21] in peripheral blood in SLE. Most of the works noted that in SLE, there were increased NETs: increased formation of NETs by neutrophils, decreased degradation of NETs by serum, or increased levels of indirect markers of NETs in the blood. Data obtained in the present study and those obtained by other authors suggest the importance of NETs in the pathogenesis of SLE. However, there are conflicting data on the relationship between NETs and the clinical and laboratory manifestations of SLE.

Lupus nephritis is one of the most severe and prognostically unfavorable manifestations of SLE. The pathogenesis of renal damage is associated with the deposition of immune complexes, which leads to the activation of the complement system and the attraction of inflammatory cells, including neutrophils, to the renal glomeruli [22]. It was found that patients with SLE and glomerulonephritis had significantly higher serum MPO-DNA complex levels (Figure 5). The data obtained partially agree with the results of other researchers. As in the present work, Bruschi et al. [14] found that in patients with SLE, the levels of the MPO-DNA complex were significantly higher in patients with lupus nephritis compared to those without renal involvement. Hakkim et al. [17] studied the ability of the serum of SLE patients to degrade NETs. They found that patients with SLE who had reduced NETs elimination and, accordingly, higher levels of NETs in the blood were more likely to have lupus nephritis. Interestingly, the authors found deposition of NETs in the renal glomeruli in an SLE patient with decreased NETs degradation. Leffler et al. [18] also found an association between impaired NETs elimination and lupus nephritis in SLE. Moreover, class IV lupus nephritis was significantly more frequent in the group of patients with decreased NETs degradation. A prospective study by Leffler et al. [19] demonstrated that a reduced NET degradation ability in SLE patients was associated with the presence of markers of active lupus nephritis, such as proteinuria, cellular casts, and leukocyturia. Zhang et al. [20] studied nonspecific markers of NETs in SLE. The authors found that the levels of circulated cell-free DNA (cfDNA) were significantly higher in patients with lupus nephritis compared to those without renal damage. The levels of cfDNA were particularly high in patients with active lupus nephritis, which correlated directly with daily proteinuria and inversely with endogenous creatinine clearance. Although the study by El-Ghoneimy et al. [16] found no significant difference in NETs levels between patients with and without lupus nephritis, patients with a higher renal SLEDAI had significantly higher NETs levels that correlated with the levels of daily proteinuria. There are also opposite results regarding NETs and lupus nephritis in SLE [13,15]. Van der Linden et al. [15] studied the ability of neutrophils to produce NETs under the influence of plasma of SLE patients. The authors found no correlation between the increased release of NETs and lupus nephritis. The study by Hanata et al. [13] also found no relationship between the MPO-DNA complex and lupus nephritis. Hanata et al. suggested that their results were related to the features of the cohort they studied: most patients had the “inflammatory” phenotype of SLE with fever; serositis; arthritis; myositis; and, less often, lupus nephritis. In the present work, 48.1% of patients with SLE had lupus nephritis, more in line with patient characteristics in other similar studies [14,17,18,20], where there was a sufficient number of patients with lupus nephritis and an association between renal disease and elevated NETs levels. Villanueva et al. [22] conducted an interesting study. They analyzed renal biopsy specimens from nine SLE patients with lupus nephritis. In the morphological study of the kidneys, NETs were found in 67% of cases. Moreover, the percentage of glomeruli infiltrated with NETs was higher in patients with grade IV lupus nephritis and/or a higher biopsy activity index. In addition, patients with NETs in the glomeruli had, on average, higher serum levels of antibodies to dsDNA than patients without NETs in the glomeruli.

DsDNA antibody positivity is a risk factor for lupus nephritis [23] and increased NETs formation by neutrophils in SLE [24]. It is known that antibodies to dsDNA and the complement system can contribute to the formation of NETs; in turn, the components of NETs themselves can activate the complement system and stimulate the synthesis of antibodies to dsDNA, which creates a vicious cycle of inflammation in SLE [25]. However, data on the relationship between NETs and immunological markers of SLE remain inconsistent. The present study found an association between positive levels of the MPO-DNA complex and dsDNA antibody positivity and hypocomplementemia. In addition, a direct correlation was found between the MPO-DNA complex and antibodies to dsDNA and an inverse correlation with complement components C3 and C4. Almost all researchers found an association between NETs and antibodies to dsDNA [15,16,17,18,19,21], and some found an association between NETs and hypocomplementemia [16,18,19]. Other authors [20] found no correlation between NETs and immunological markers of SLE; in turn, Hanata et al. [13] found a negative correlation between the MPO-DNA complex and levels of antibodies to dsDNA, which may be due to the specificity of the cohort they studied.

The data on the correlation between the markers of NETosis and SLE activity are also contradictory. In this study, it was found that patients with high SLE activity were more likely to be positive for the MPO-DNA complex. The chance of having high SLE activity with positive levels of the MPO-DNA complex was 3.2 times higher than with negative levels (Table 1). In addition, the MPO-DNA complex was positively correlated with SLEADI-2K (Figure 4a). The obtained results are consistent with those of Leffler et al. [18]. In their other work [19], they demonstrated that a reduced ability to degrade NETs was associated with active SLE, in particular with the presence of active lupus nephritis. El-Ghoneimy et al. [16] also found a positive correlation between NETs and SLEDAI-2K. On the other hand, other authors found no significant association between NETs and disease activity in SLE [13,14,15,20,21]. The results obtained may be related to the heterogeneity of the patient samples studied as well as to different methods of determining NETs.

Table 5 summarizes the literature data on the association between NETs and SLE activity, lupus nephritis, and immunological markers of SLE.

Nucleosomes are another potential marker of NETosis. According to different authors [26,27,28], nucleosome levels are elevated in SLE patients, which may be associated with increased lymphocyte apoptosis or NETosis. However, in the present work, the opposite results were obtained. Significantly, lower levels of nucleosomes were observed in patients with SLE compared to healthy controls. Moreover, in this work, lower nucleosome values were associated with high SLE activity, the presence of glomerulonephritis, and arthritis.

We expected to find an increase in the levels of nucleosomes, as well as the MPO-DNA complex, suggesting that both indicators are markers of NETosis. However, opposite results were obtained. One possible explanation for the low levels of nucleosomes in patients is the possible presence of an inhibitor of nucleosome release in the serum of SLE patients, as was found in a study by Marsman et al. [29]. It is possible that the use of nucleosomes as markers of NETosis is associated with great difficulties, given the influence of many factors on their levels. Further study of the role of nucleosomes in the pathogenesis of SLE and as markers of NETosis is required.

According to several studies [30,31,32,33], NETs are involved in the pathogenesis of APS. In this work, no significant differences were found between the levels of the MPO-DNA complex in patients with PAPS, SLE with APS, and healthy controls. At the same time, positive values of the MPO-DNA complex were found in 25% (22 of 88) of patients with APS. No significant associations were found between the MPO-DNA complex and thrombosis, obstetric pathology, antiphospholipid antibody profile, or levels of positivity. Patients with highly positive levels of aCL IgM and anti-β2GP1 IgM were significantly less likely to have decreased serum nucleosome levels compared with patients with lower or negative levels of these antibodies. Other clinical and laboratory manifestations of APS were not significantly associated with low nucleosome levels. The inconsistent results that were obtained may be due to a long post-thrombotic period and time after an obstetric pathology, as well as almost all patients received long-term antiplatelet drugs, hydroxychloroquine, which could affect the ability of neutrophils to produce NETs in APS.

## 4. Materials and Methods

A total of 138 people were included in the cross-sectional study: 30 with SLE without APS, 47 with SLE and APS, 41 patients with primary antiphospholipid syndrome (PAPS), and 20 practically healthy individuals without a history of oncological diseases or acute infectious diseases at the time of blood sampling. Written informed consent was obtained from all participants. The characteristics of patients and healthy controls are presented in Appendix A. Patients and healthy controls were comparable in gender. Patients with PAPS and SLE with APS were older than patients with SLE without APS and healthy donors. Patients with SLE without APS and PAPS had a shorter disease duration compared with patients with SLE with APS (*p* < 0.0001 and *p* = 0.030, respectively).

The diagnosis of SLE was based on the 2012 Systemic Lupus International Collaborating Clinics classification criteria for SLE (SLICC) [34] and the 1997 American College of Rheumatology (ACR) classification criteria [35] (Appendix A). Patients with SLE without APS had more frequent lupus glomerulonephritis than patients with SLE with APS (*p* = 0.032). Other clinical manifestations in both groups of SLE were comparable in frequency. SLE activity was assessed using the Systemic Lupus Erythematosus Disease Activity Score Index 2000 (SLEDAI-2K) [36]. At the time of inclusion in the study, SLE patients without APS had higher SLEDAI-2K activity than those with SLE with APS (*p* < 0.0001). Irreversible organ damage was assessed using the SLICC/ACR damage index (The Systemic Lupus International Collaborating Clinics/American College of Rheumatology (SLICC/ACR) Damage Index) [37]. The SLICC/ACR damage index was significantly higher in patients with SLE and APS (*p* < 0.0001).

The diagnosis of APS was based on the 2006 international classification criteria [38] (Appendix A). PAPS was verified in a patient in the absence of signs of any other disease and the presence of those of definite APS

Recruitment of patients for clinical and laboratory tests was carried out at the V.A. Nasonova Research Institute of Rheumatology. All patients underwent a standard examination, which included a chest X-ray, electrocardiogram, echocardiogram, and general clinical methods of blood and urine examination. The determination of antibodies to double-chiral deoxyribonucleic acid (anti-dsDNA), antibodies to Sm-antigen (aSm), antibodies to cardiolipin of immunoglobulins G and M (IgG/IgM aCL), and antibodies to beta-2 glycoprotein 1 of immunoglobulins G and M (IgG/IgM anti-β2P1) was carried out via an enzyme immunoassay (ELISA) on an automatic analyzer for laboratory diagnostics of autoimmune diseases, Alegria (Orgentec Diagnostika GmbH, Mainz, Germany), with a set of reagents for the determination of antibodies from Orgentec Diagnostika GmbH, Mainz, Germany. Anti-dsDNA and aSm were measured in IU/mL. IgG aCLs were measured in the phospholipid-binding activity of IgG aCLs per 1 IU/mL in GPL units, and IgM aCLs were measured in the phospholipid-binding activity of IgM aCLs per 1 IU/mL in MPL units. IgG/IgM anti- β_2_GP1 was measured in IU/mL. Values >25.00 GPL for IgG aCL, >24.70 MPL for IgM aCL, >15.30 IU/mL for IgG anti-B_2_GP1, and >17.00 IU/mL for IgM anti-B_2_GP1 were considered positive [39]. Antinuclear antibodies were determined using indirect immunofluorescence using HEp-2 cells (epithelial cells of human laryngeal cancer) as a substrate. The concentration of complement components C3 and C4 was determined by immunonephelometry on a BN ProSpec analyzer (Siemens, Marburg City, Germany) using Siemens reagent kits. The units of measurement were grams per liter (g/L). Values of less than 0.900 g/L were taken for a reduced level of the C3 component of the complement, and values of less than 0.100 g/L were taken for C4. The study of lupus anticoagulant (LA) was carried out on an automatic coagulometer from Siemens Healthcare (Erlangen City, Germany). The LA study was conducted on patients who did not receive anticoagulants.

### 4.1. Serum Histone-Associated-DNA-Fragments Immunoanalysis (Nucleosomes)

The streptavidin-coated 96-well microplates (Roche, Cell Death Detection ELISA PLUS, Basel, Switzerland) were washed once with PBST buffer. Randomized serum samples were diluted ten times with incubation buffer (Roche, Cell Death Detection ELISA PLUS) supplemented with biotinylated anti-histone antibody (clone H11-4, Roche, Cell Death Detection ELISA PLUS), diluted according to manufacturer’s instructions, subjected to plates, and incubated overnight at 4 °C. Each serum sample was analyzed in duplicate on different plates. Next, plates were washed by PBST three times, and peroxidase-conjugated Anti-DNA-POD monoclonal antibody (clone MCA-33, Roche, Cell Death Detection ELISA PLUS) was added to wells in dilution recommended by manufacturer. Plates were washed five times by PBST, and 50 μL of TMB was added to each well. Reaction was terminated by 50 μL of 10% phosphoric acid. The optical density (OD) of each well was subsequently measured at a wavelength of 450 nm (OD_450_), with 630 nm used as a reference correction utilizing CLARIOstar plate reader (BMG LABTECH, Ortenberg, Germany). Serial dilutions of the DNA-Histone-Complex standard (Roche, Cell Death Detection ELISA PLUS) were used as a positive control (1:2, 1:4, 1:8, 1:16, 1:32, 1:64, 1:128, and 1:333) on each 96-well plate. Individual values from each plate (n = 4), as well as average and standard deviation, are shown in Figure 10a. The reference values corresponding to the 5th percentile and 95th percentile of healthy controls were 0.1528–1.0442 OD_450_. One of the healthy controls had low levels of nucleosomes (0 and 1515 OD_450_).

### 4.2. Serum NETs Immunoanalysis (MPO-DNA Complex)

The MaxiSorp 96-well microplates (Nunc, Rochester, NY, USA) were coated overnight with 5 μg/mL anti-myeloperoxidase antibody (clone 07-496-I, Merck, Darmstadt, Germany) in carbonate-bicarbonate buffer, pH 9.6. Plates were washed once with PBST buffer (PBS supplemented with 0.1% Tween-20) and further blocked with 3% BSA for 1 h at room temperature. Plates were washed by PBST three times. Randomized serum samples were diluted ten times with incubation buffer (Roche, Cell Death Detection ELISA PLUS), subjected to plates, and incubated for 2 h at room temperature. Each serum sample was analyzed in duplicate on different plates. Next, plates were washed by PBST three times, and 50 μL of peroxidase-conjugated Anti-DNA-POD monoclonal antibody (clone MCA-33, Roche, Cell Death Detection ELISA PLUS) was added to wells in dilution recommended by manufacturer. Plates were washed five times by PBST, and 50 μL of 3,3′,5,5′-tetramethylbenzidine (TMB) was added to each well. Reaction was terminated by 50 μL of 10% phosphoric acid. The optical density (OD) of each well was subsequently measured at a wavelength of 450 nm (OD_450_), with 630 nm used as a reference correction by CLARIOstar plate reader. Serial dilutions of the DNA-Histone-Complex standard (Roche, Cell Death Detection ELISA PLUS) were used as a positive control (1:2, 1:4, 1:8, 1:16, 1:32, 1:64, 1:128, 1:333) on each 96-well plate. Individual values from each plate (n = 4), as well as average and standard deviation, are shown in Figure 10b. The reference values corresponding to the 5th percentile and 95th percentile of healthy controls were 0.0292-0.09335 OD_450_. One of the healthy controls tested positive for the MPO-DNA complex (0.094 OD_450_).

### 4.3. Statistics

Quantitative variables were described as M ± σ, where M is mean; σ is standard deviation; and Me (Q25; Q75), where Me is median and Q25 and Q75 are 25% and 75% percentiles, respectively. Distributions were checked for normality using the Shapiro–Wilk test. Quantitative variables were compared using the Mann–Whitney test and the Kruskal–Wallis test adjusted with the Bonferroni correction. Correlation analysis was performed using Spearman’s rank-order correlation coefficient. For the comparison of qualitative variables, χ^2^ (Pearson test) was used; Fisher’s criterion was used when the number of cases was less than 10. ROC analysis (ROC receiver operating characteristic) was used to determine AUC (Area under the Curve), sensitivity (Se), and specificity (Sp). Differences were considered to be statistically significant at *p* < 0.05.

## 5. Conclusions

This study revealed an increase in a specific marker of NETosis, the MPO-DNA complex, in the serum of SLE patients but not APS. MPO-DNA complex positivity was associated with the presence of lupus nephritis, antibodies to dsDNA, hypocomplementemia, and overall SLE activity, indicating the relationship between NETosis and the clinical and immunological manifestations of SLE. Elevated levels of the MPO-DNA complex can be regarded as a promising biomarker of lupus nephritis, disease activity, and immunological disorders in SLE patients. Significantly lower levels of nucleosomes were found in patients with SLE and SLE with APS. Low values were associated with the presence of lupus nephritis, arthritis, and high SLE activity.

## Figures and Tables

**Figure 1 ijms-24-09210-f001:**
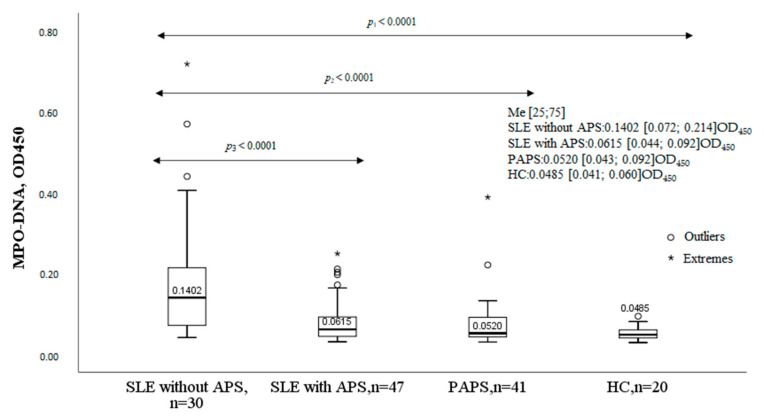
MPO-DNA complex in SLE, SLE with APS, PAPS, and HC. Note: SLE, systemic lupus erythematosus; APS, antiphospholipid syndrome; PAPS, primary antiphospholipid syndrome; HC, healthy controls; MPO-DNA complex, myeloperoxidase–deoxyribonucleic acid complex; OD, optical density; Me, median with an interquartile range; *p*, probability.

**Figure 2 ijms-24-09210-f002:**
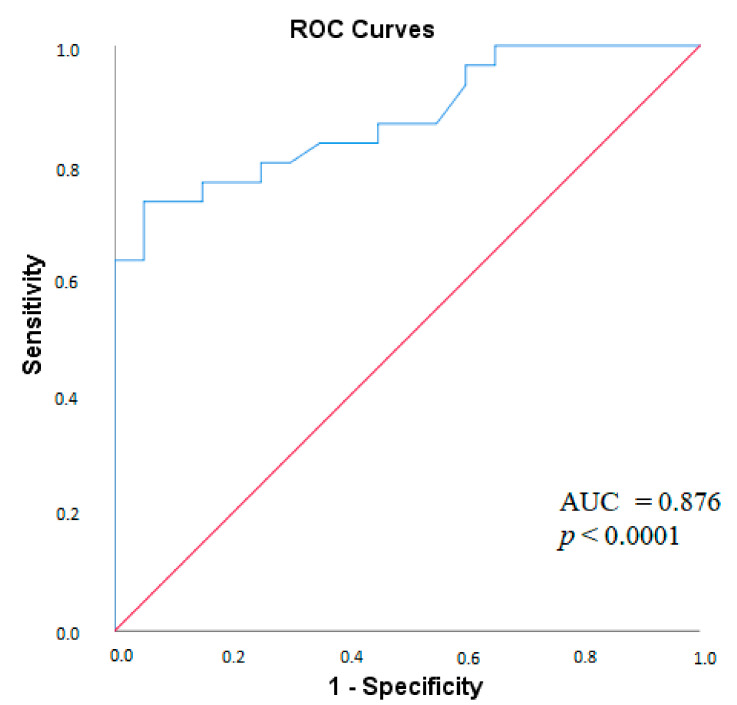
ROC-curve of the MPO-DNA complex for the diagnosis of SLE without APS. Note: ROC, receiver operating characteristic, MPO-DNA complex, myeloperoxidase-deoxyribonucleic acid complex, SLE, systemic lupus erythematosus; AUC, area under curve; *p*, probability.

**Figure 3 ijms-24-09210-f003:**
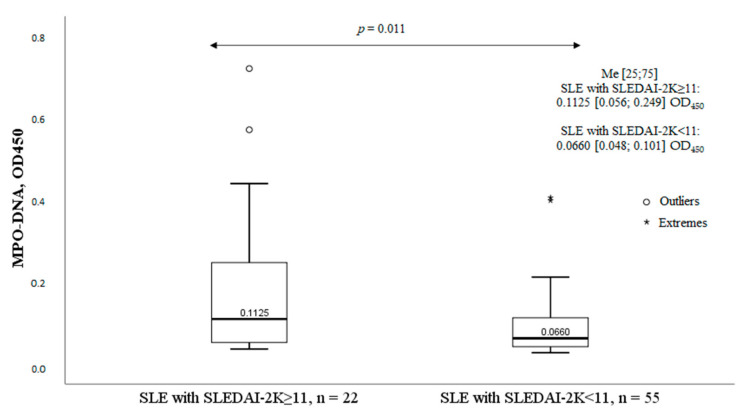
MPO-DNA complex levels and SLE activity. Note: MPO-DNA complex, myeloperoxidase-deoxyribonucleic acid complex; OD, optical density; SLE, systemic lupus erythematosus; Me, median with an interquartile range; *p*, probability; SLEDAI-2K, Systemic Lupus Erythematosus Disease Activity Index; HC, healthy controls.

**Figure 4 ijms-24-09210-f004:**
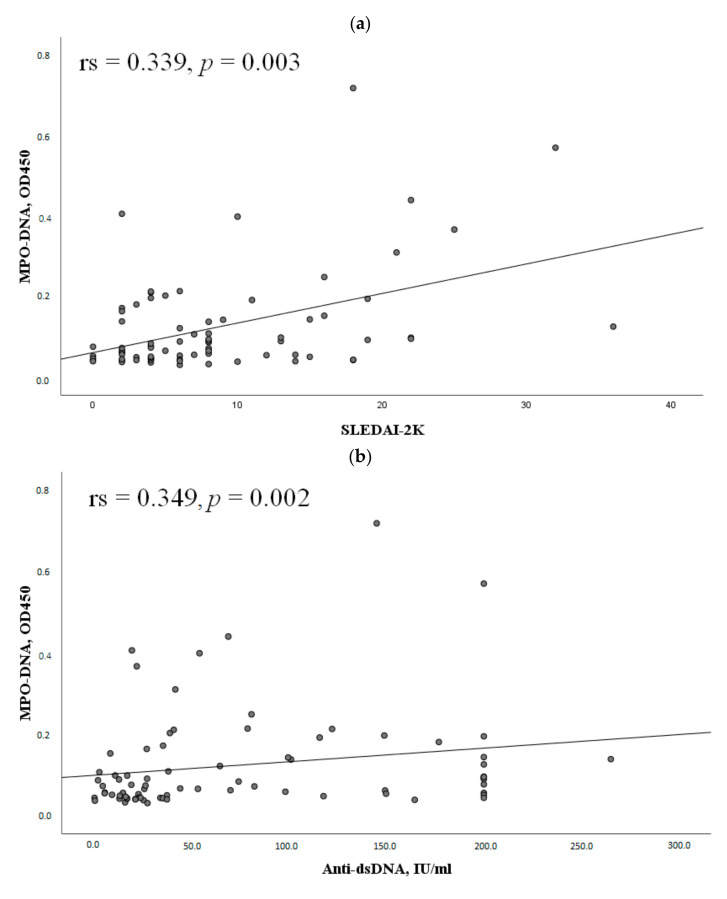
Correlations of the MPO-DNA complex levels with some clinical and laboratory manifestations of SLE. Note: (**a**) Correlation of the MPO-DNA complex with the SLEDAI-2K index (n = 77); (**b**) Correlation of the MPO-DNA complex with antibodies to dsDNA (n = 77); (**c**) Correlation of the MPO-DNA complex with C3 (n = 77); (**d**) Correlation of the MPO-DNA complex with C4 (n = 77). MPO-DNA complex, myeloperoxidase-deoxyribonucleic acid complex; SLE, systemic lupus erythematosus; SLEDAI-2K, Systemic Lupus Erythematosus Disease Activity Index; anti-dsDNA, anti-double-stranded DNA; rs, Spearman’s rank correlation coefficient; *p*, probability.

**Figure 5 ijms-24-09210-f005:**
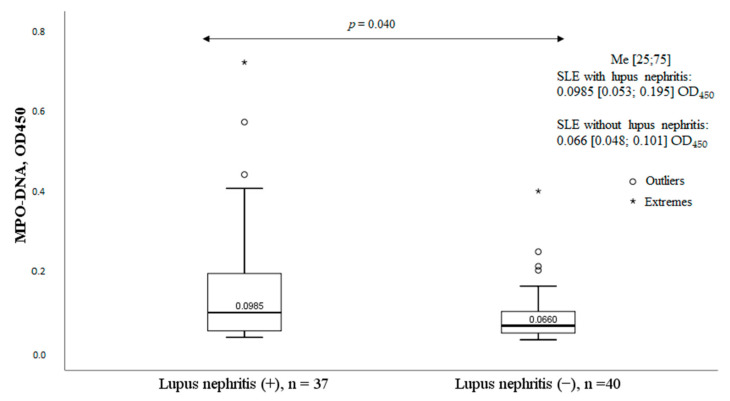
Levels of the MPO-DNA complex in patients with and without lupus nephritis. Note: MPO-DNA complex, myeloperoxidase-deoxyribonucleic acid complex; OD, optical density; SLE, systemic lupus erythematosus; Me, median with an interquartile range; *p*, probability; HC, healthy controls.

**Figure 6 ijms-24-09210-f006:**
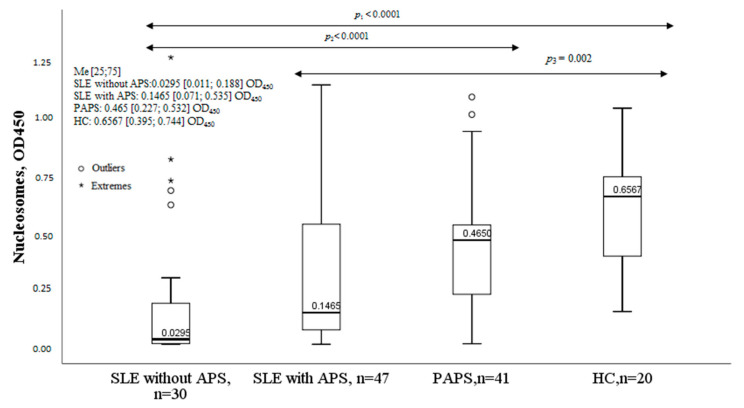
Nucleosomes in SLE, SLE with APS, PAPS, and HC. Note: SLE, systemic lupus erythematosus; APS, antiphospholipid syndrome; PAPS, primary antiphospholipid syndrome; HC, healthy controls; OD, optical density; Me, median with an interquartile range; *p*, probability.

**Figure 7 ijms-24-09210-f007:**
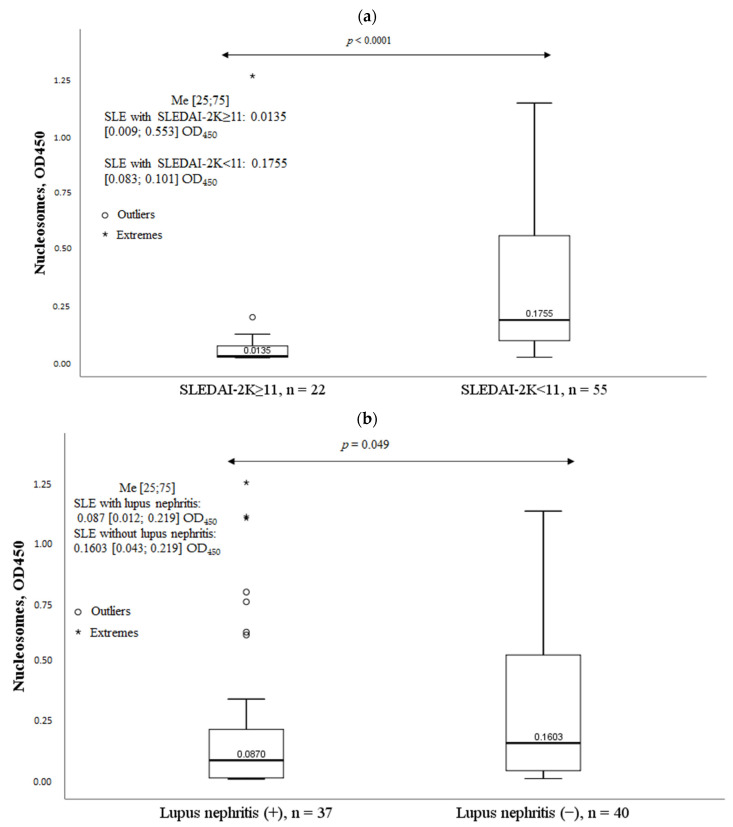
Nucleosomes in some clinical manifestations of systemic lupus erythematosus. Note: (**a**). Nucleosomes according to SLE activity; (**b**). Nucleosomes in lupus nephritis; (**c**). Nucleosomes in arthritis; SLEDAI-2K, Systemic Lupus Erythematosus Disease Activity Index; OD, optical density; Me, median with an interquartile range; *p*, probability.

**Figure 8 ijms-24-09210-f008:**
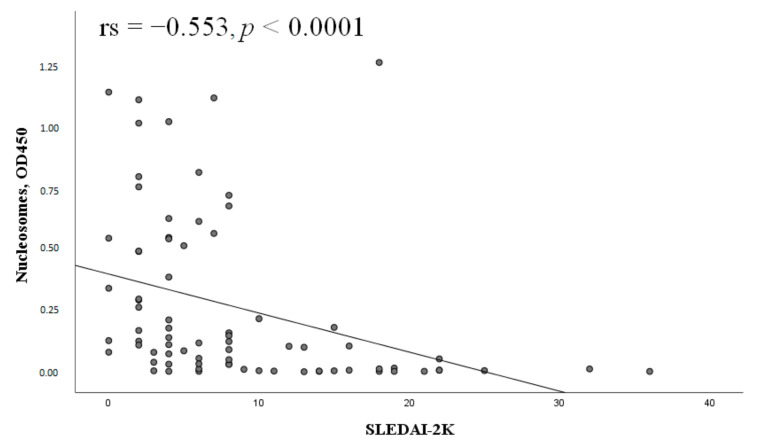
Correlation of nucleosomes with the SLEDAI-2K index (n = 77). Note: SLEDAI-2K, Systemic Lupus Erythematosus Disease Activity Index; OD, optical density; rs, Spearman’s rank correlation coefficient; *p*, probability.

**Figure 9 ijms-24-09210-f009:**
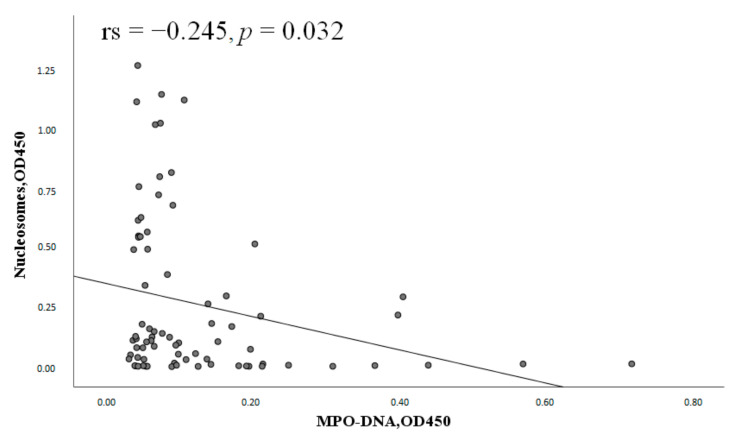
Correlation of nucleosomes with the MPO-DNA complex (n = 77). Note: MPO-DNA complex, myeloperoxidase–deoxyribonucleic acid complex; OD, optical density; rs, Spearman’s rank correlation coefficient; *p*, probability.

**Figure 10 ijms-24-09210-f010:**
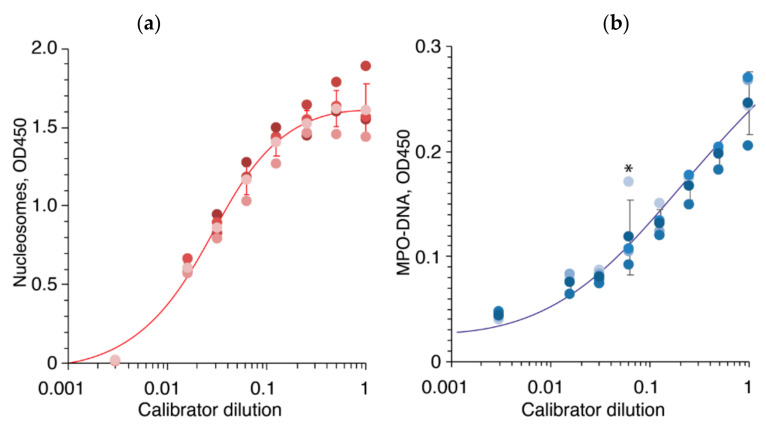
ELISA calibration curves of the nucleosomes (**a**) and MPO-DNA (**b**) are shown by red and blue lines, respectively. Serial dilutions of the DNA-Histone-Complex standard (Roche, Cell Death Detection ELISA PLUS) were used as a positive control (1:2, 1:4, 1:8, 1:16, 1:32, 1:64, 1:128, 1:333) on each 96-well plate. Individual values from each plate (n = 4), as well as average and standard deviation, are shown by dots and bars, respectively. Asterisk denotes outlier value.

**Table 1 ijms-24-09210-t001:** The frequency of detection of the MPO-DNA complex in patients with SLE depending on clinical manifestations.

Parameter		MPO-DNA Complex (+), n = 30, n (%)	MPO-DNA Complex (-), n = 47, n (%)	χ^2^; *p* OR and 95% CI
Lupus nephritis	yes	20 (67)	17 (36)	6.82; 0.009 3.53 [1.35–9.26]
	no	10 (33)	30 (64)	
Positive antibodies to dsDNA	yes	26 (87)	30 (64)	4.82; 0.036 3.68 [1.09–12.34]
	no	4 (13)	17 (36)	
Hypocomplementemia	yes	23 (77)	22 (47)	6.72; 0.01 3.73 [1.34–10.37]
	no	7 (23)	25 (53)	
High SLE activity	yes	13 (43)	9 (19)	5.25; 0.037 3.23 [1.16–8.99]
	*no*	17 (57)	38 (81)	

Note: MPO-DNA complex, myeloperoxidase–deoxyribonucleic acid complex; χ^2^, agreement criterion; *p*, probability; OR, odds ratio; CI, confidence interval; anti-dsDNA, anti-double-stranded DNA; SLE, systemic lupus erythematosus.

**Table 2 ijms-24-09210-t002:** Levels of the MPO-DNA complex depending on the presence of immunological indicators of SLE (n = 77).

Parameters		MPO-DNA Complex Levels, OD_450_ Me, [25;75]	*p*
Positive anti-dsDNA	yes (n = 56)	0.090 [0.0518;0.186]	*p* = 0.037
	no (n = 21)	0.056 [0.0435;0.089]	
Hypocomplementemia	yes (n = 45)	0.095 [0.054;0.197]	*p* = 0.032
	no (n = 32)	0.065 [0.044;0.095]	
Positive anti-Sm antibody	yes (n = 9)	0.089 [0.0655;0.181]	*p* = 0.706
	no (n = 42)	0.089 [0.050;0.153]	
Positive aPL	yes (n = 35)	0.063 [0.044;0.142]	*p* = 0.142
	no (n = 42)	0.091 [0.055;0.143]	

Note: MPO-DNA complex, myeloperoxidase–deoxyribonucleic acid complex; OD, optical density; antibodies to dsDNA, antibodies to double-stranded DNA; antibodies to Sm, antibodies to the Smith antigen; aPL, antiphospholipid antibodies.

**Table 3 ijms-24-09210-t003:** Frequency of detecting low and normal nucleosome values depending on the clinical manifestations of SLE.

Parameters		Nucleosomes (−), n = 45, n (%)	Nucleosomes (+), n = 32, n (%)	χ^2^; *p* OR and 95% CI
Lupus nephritis	yes	26 (57.8)	11 (34.4)	4.1; 0.043 2.61 [1.02–6.68]
	no	19 (42.2)	21 (65.6)	
Arthritis	yes	36 (80)	19 (59.4)	3.89; 0.048 1.71 [1.03–2.83]
	no	9 (20)	13 (40.6)	
High SLE activity	yes	20 (44.4)	2 (6.25)	13.37; <0.0001 12.0 [2.55–56.39]
	no	25 (55.6)	30 (93.8)	
Positive anti-dsDNA	yes	36 (80)	20 (62.5)	2.89; 0.076 2.4 [0.86–6.67]
	no	9 (20)	12 (37.5)	
Hypocomplementemia	yes	29 (64.4)	16 (50)	1.61; 0.205 1.81 [0.72–4.56]
	no	16 (35.6)	16 (50)	

Note: n, number of patients; χ^2^, agreement criterion; *p*, probability; OR, odds ratio; CI, confidence interval; dsDNA, double-stranded DNA; SLE, systemic lupus erythematosus; anti-dsDNA, antibodies to double-stranded DNA.

**Table 4 ijms-24-09210-t004:** Frequency of nucleosome detection in APS with highly positive IgM aCL and IgM aβ2GP1.

Parameter		Nucleosomes (−), n = 32, n (%)	Nucleosomes (+), n = 56, n (%)	χ^2^; *p* OR and 95% CI
Highly positive levels of IgM aCL	yes	1 (3)	11 (20)	4.52; 0.049 0.14 [0.017–1.112]
	no	31 (97)	45 (80)	
Highly positive levels of IgM aβ2GP1	yes	0	49 (88)	4.21; 0.047 0.61 [0.52–0.73]
	no	32 (100)	7 (12)	

Note: n, number of patients; χ^2^, agreement criterion; *p*, probability; OR, odds ratio; CI, confidence interval; dsDNA, double-stranded DNA; aCL, anti-cardiolipin antibodies; IgM, immunoglobulin M; aβ2GP1, antibodies to β2 glycoprotein 1.

**Table 5 ijms-24-09210-t005:** NETs and SLE activity, lupus nephritis, and immunological markers of SLE.

Authors	Number of Patients	Methods of NETs Detection	Association of NETs with Lupus Nephritis	Association of NETs with Daily Proteinuria	Association of NETs with Anti-dsDNA	Association of NETs with C3, C4	Association of NETs with SLE Activity
Hakkim et al., 2010 [17]	SLE 61, RA 30, HC 54	Destruction of NETs by serum	+	ND	+	ND	ND
Leffler et al., 2012 [18]	SLE 94, HC 54	1. Destruction of NETs by serum 2.DNAase activity	+	ND	+	+	+
Leffler et al., 2013 [19]	SLE 69	Destruction of NETs by serum	+	ND	+	+	+
Zhang et al., 2014 [20]	SLE 54, HC 43	Plasma cfDNA levels	+	+	−	−	−
van der Linden et al., 2018 [15]	SLE 55, SLE with APS 38, PAPS 28, HC 27	NETs production	−	ND	+	−	−
El-Ghoneimy et al., 2019 [16]	SLE 50, HC 50	NETs production	+	+	+	+	+
Jeremic et al., 2019 [21]	SLE 111, HC 50	cfDNA levels	−	ND	+	−	−
Bruschi et al., 2020 [14]	SLE 216, SLE with lupus nephritis 103, HC 50	1.MPO-DNA complex ELISA 2.NETs production 3.DNAase activity	+	ND	ND	−	−
Hanata et al., 2022 [13]	SLE 33, HC 19	MPO-DNA complex ELISA	−	ND	+	-	−
Reshetnyak et al., 2023 (current study)	SLE 30, SLE + APS 47, PAPS 41, HC 20	MPO-DNA complex ELISA	+	-	+	+	+

Note: SLE, systemic lupus erythematosus; MPO-DNA complex, myeloperoxidase–deoxyribonucleic acid complex; ELISA, enzyme immunoassay; NETs, neutrophil extracellular traps; RA, rheumatoid arthritis; cfDNA, cell-free DNA; SLEDAI, Systemic Lupus Erythematosus Disease Activity Index; APS, antiphospholipid syndrome; PAPS, primary antiphospholipid syndrome; ND, no data**.**

## Data Availability

The authors confirm that the data supporting the findings of this study are available within the article and its Appendix A).

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
