# Peer review of "Markers of NETosis in Patients with Systemic Lupus Erythematosus and Antiphospholipid Syndrome"

_ijms, 2023, doi:10.3390/ijms24119210_

Round 1

Reviewer 1 Report

Presented is a manuscript evaluating NETosis parameters as markers for SLE and APS. This is quite relevant and really interesting, because markers for SLE diagnosis and activity are needed.

The paper is difficult to understand. Starting in the abstract: There is no introduction. For what should MPO-DNA be clinical significant? For diagnosis or for therapy/activity? Which methods were used? Was the study approved by an ethics committee? Informed consent?

In the introduction, it does not become clear, what the aim of the study is. The methods are not mentioned in the abstract, introduction or results. I figured out, that ELISAs were performed, when I read the methods section.

These ELISAs are the main problem of this study: Presented are raw optical density values of the photometer. Apparently, no controls were used like positive/negative control or standard concentrations. Therefore, it is simply not good laboratory practise to compare values which were obtained on different 96 well plates. If for example Lupus Nephritis patients were placed on one plate and patients withough LN on another, the differences in figure 5 can simply be explained by different incubation times or temperature differences. Without a proper control, the data cannot be used.

The rest of the paper also is not very conclusive and difficult to read. Some examples:

SLE patients are divided into two groups (line 93) according to their MPO-DNA positivity (where is the cutoff for positivity? how is it defined?). Later, patients are divided depending on SLEDAI-score (line 110). It does not become clear, which group is used when and why.

In the discussion (and the abstract), significant differences are stated. In the conclusion, just non-significant data is mentioned.

The message, that higher MPO-DNA values are correlated with higher SLE activity, is interesting and important, but I am not sure if we can trust this data. This message is also not presented clearly and not in the conclusion.

I think this is a nice cohort. It is described quite well in the methods section. Unfortunately, there are major flaws in the study design as well as in the presentation of the data.

language quality needs to be improved. Some sentences are too long and not understandable.

Author Response

Dear reviewer! Thank you very much for reviewing our article. We tried to take into account your comments. Thank you for the work done, for your valuable comments and suggestions! We have attached our answers below.

Dear Reviewer! Once again, we thank you for your work and are reattaching our response due to revisions to the text.

Reviewer 2 Report

 The paper by Reshetnyak TM et al. is focused on the possible clinical significance of the circulating levels of MPO-DNA complexes derived from the netotic process and nucleosomes in patients with SLE and APS. The authors conclude that MPO-DNA complexes correlate positively with disease activity in SLE and with some routine clinical parameters, while the opposite is true for nucleosome levels. Furthermore, there appears to be no correlation between levels of the complexes and clinical manifestations in patients with APS.

In the light of the contradictory results existing in the Literature, the topic is certainly of interest. The data in the main text are clearly described as well as the methodology used, and the text is therefore fluent. However, the authors should spend a few more lines commenting/speculating why the manifestation of APS (both primary and in Lupus patients) seems to be associated with the decrease of circulating MPO-DNA levels. Furthermore, the authors could speculate why there are fewer nucleosomes in patients with high MPO-DNA complexes (assuming both factors result from netosis).

Moreover, authors should add sample size to each figure legend.

Author Response

Dear reviewer! Thank you very much for your feedback!  We have tried to take into account your comments. Thank you for the work you have done, for your valuable comments and suggestions regarding our research paper!

Reviewer 3 Report

1.  General comments:

A. There is considerable redundancy or unnecessary repetition in describing p-values and in defining abbreviations in the text, figures and figure legends. For example Fig. 1 legend should be shortened, since the p-values are provided in the figure itself.

B. “p-value” normally refers to statistical probability, not “reliability” (whatever that means).

C.  It would seem that the data in Tables 1 and 2 could be combined.

2. Based on Fig. 1 all SLE patients and all normals had positive MPO-DNA activity, so what is meant by or defined as  “negative MPO-DNA” in lines 94-95 and in Table 1 is unclear. 

3. Table 3:  The criteria for low and high levels of nucleosomes need to be defined.

4.  Based on  the data in Table 3 and Fig. 6 and instead of Fig. 7a, it would seem of interest to graphically compare nucleosome levels with disease activity in SLE, analogous to what was done in Fig. 4a.

5. Line 191 should be Table 4, not Table 7.

6.  Table 6

A. The p-values do not indicate what two patient groups are being compared.

B.  For gender line, female/male ratio would be sufficient.

7.  Table 7.   SLEDAI-2K and ID SLICC/ACR disease indexes: It is unclear what the numbers in brackets refer to.

8.  Table 8. This table does not seem to be referred to in Results, although the “Conclusion” section on p. 12, lines 313-326 seems to be referring to some of this data.  Overall, the purpose or value of this table is unclear. Also, the three numbers under thrombosis are not explained.

9.  Assay for serum NETs (MPO-DNA complex):  the volumes of anti-MPO and of 1/10 diluted serum are not stated.  Volumes were also omitted for “serum histone-associated-DNA-fragments immunoanalysis (Nucleosomes)”.

10.  It might be informative to compare MPO-DNA levels with nucleosome levels considering that the MPO-DNA complexes are likely to include histone bound to the DNA.

11.  Table 5:  It would seem appropriate to include the current study in this table.

12.  There are inappropriately two sections labeled as Conclusions.

The English writing is largely fine, but the manuscript preparation seems strangely careless regarding too many or too few details.

Author Response

Dear reviewer! Thank you very much for reviewing our article. We have tried to take into account your comments. Thank you for the work you have done, for your valuable comments and suggestions, which helped us to improve the quality of our article.

Reviewer 4 Report

According to several studies , NETs are involved in the pathogenesis of APS. In this work, no significant differences were found between the levels of the MPO-DNA complex in patients with PAPS, SLE with APS, and healthy controls.

At the same time, positive values of the MPO-DNA complex were found in 25% (22 of 88) of patients with APS.

No significant associations were found between the MPODNA complex and thrombosis, obstetric pathology, antiphospholipid antibody profile, or levels of positivity.

Patients with highly positive levels of aCL IgM and anti-β2GP1 IgM were significantly less likely to have decreased serum nucleosome levels compared with patients with lower or negative levels of these antibodies.

Other clinical and laboratory manifestations of APS were not significantly associated with low nucleosome levels.

The inconsistent results that were obtained may be due to a long post-thrombotic period and time after an obstetric pathology, as well as almost all patients, received long-term antiplatelet drugs, hydroxychloroquine, which could affect the ability of neutrophils to produce NETs in APS.

According to several studies , NETs are involved in the pathogenesis of APS. In this work, no significant differences were found between the levels of the MPO-DNA complex in patients with PAPS, SLE with APS, and healthy controls.

At the same time, positive values of the MPO-DNA complex were found in 25% (22 of 88) of patients with APS.

No significant associations were found between the MPODNA complex and thrombosis, obstetric pathology, antiphospholipid antibody profile, or levels of positivity.

Patients with highly positive levels of aCL IgM and anti-β2GP1 IgM were significantly less likely to have decreased serum nucleosome levels compared with patients with lower or negative levels of these antibodies.

Other clinical and laboratory manifestations of APS were not significantly associated with low nucleosome levels.

The inconsistent results that were obtained may be due to a long post-thrombotic period and time after an obstetric pathology, as well as almost all patients, received long-term antiplatelet drugs, hydroxychloroquine, which could affect the ability of neutrophils to produce NETs in APS.

Author Response

Dear reviewer! Thank you for your feedback!

Round 2

Reviewer 1 Report

Dear authors,

Thank you for improving the paper. It is now better to understand. I have some final concerns I would like to adress step by step:

Table 5: Great table. Please Highlight somehow, that Reshetnyak et al is the current publication.

I would put tables 6,7,8 in a supplemental file, because it is important to provide this data to show, that your way to diagnose the patients is correct and that you have a lot of information about them. But this data is not important to understand your results and you have so many figures and tables.

Thank you that you provided data on the calibration. Can you mention it also for the MPO-DNA in the methods section and not only for nucleosomes? I think you used the same calibrators here?

And I agree with you, that the plates converge with each other.

It is also a very good information, that you measured each sample on two different plates.

I understand now, that with this standard you were not able to calculate concentrations. This is a major problem of all publications analysing MPO-DNA; Can you think of a way to include your calibrators into the calculation? For example do something like Hayden et a 2021 (PLOS ONE). They set one concentration of the calibrator to OD1 and corrected all values accordingly. Maybe that would help readers to trust the data. But that you provide the calibration curves and state that each sample was measured twice, is a big step.

Figure 10: Great that you have standard curves, but I do not understand the connection to your data and the figure legend is not very clear. For me it is not clear, what you mean with reference values and why you include it into the figure description because it is not indicated in the figure itself. Do you mean the range of the healthy control values? So everything above is „positive“?

It is also not clear, what the different colors in this figure mean.

Can you include your „reference range“, meaning the range or CI of the healthy controls into your figures? Figure, 3, 5, 7? Maybe just with a dashed line?

Thank you for resolving the issue with the two conclusions. I find it strange, that the conclusion is after the methods section, but that might be a prerequisite of the journal? I like it better when it concludes the discussion. Which your discussion does, it is just at a different place.

Author Response

Dear reviewer! Thank you very much for your valuable comments and feedback regarding our research paper. It is helped us to improve the quality of the article. We have tried to take into account your comments. Thank you again for the work you have done!

Reviewer 2 Report

The authors responded satisfactorily to all comments.

Reviewer 3 Report

Revised manuscript is substantially improved.

Round 3

Reviewer 1 Report

Thank you for all the changes.